# Research on the evaluation method of agricultural intelligent robot design solutions

Qian Tang, Ying-Wen Luo *, Xiao-Di Wu

College of Industrial Design, Hubei University of Technology, Wuhan, China

* 102001275@hbut.edu.cn

## Abstract

### Background

At present, agricultural robots are produced in large quantities and used in agricultural planting, and the traditional agricultural model is gradually shifting to rely on the Internet of Things and sensors to accurately detect crop growth information. The scientific and rational design of agricultural robots plays a huge role in planting and production efficiency, however, the factors affecting their design are complex and ambiguous, so it is necessary to use a rational evaluation system to make a preferential decision among multiple design options.

### Purposes

In order to reduce the subjectivity and blindness of program selection in the process of agricultural robot design, make the decision more objective and reasonable, and thus enhance the practicality and scientificity of the program, a new comprehensive evaluation method based on user requirements is proposed.

### Methods

First, after researching and interviewing users and farming operations, obtaining raw information on requirements, using the Kano model to classify the requirements and establishing an evaluation index system. Secondly, the combination of hierarchical analysis(AHP) and entropy weighting method is used to assign weights to the evaluation index system, calculate the weight value and importance ranking of each index, and carry out various program designs based on the ranking. Finally, the VIKOR method was applied to evaluate and rank the design solutions.

### Results

The new evaluation method can better complete the preferential decision of the agricultural robot design scheme and get a more perfect design scheme, which reduces the influence of human subjective thinking in the decision-making process.

**Data Availability Statement:** Data are within the paper and the Supporting Information file.

**Funding:** Project source unit: 2021 Hubei Provincial Education Department Philosophy and Social Science Research Project [21Q075]

**Competing interests:** The authors have declared that no competing interests exist.

## Conclusions

The method not only corrects the traditional evaluation method, but also effectively improves the accuracy and comprehensiveness of the design evaluation process. It also provides a reference for designers to preferably select design solutions and promotes the development of small mobile machines in the context of smart agriculture.

## 1.Introduction

With the development of information technology, the Internet of Things (IoT) has started to be used extensively, which has greatly enhanced the wisdom of society. Sensors, robots, and communication devices relying on IoT technology are gradually being maturely applied to various fields [1], including agricultural systems. The application of IOT in agriculture is mainly reflected in agricultural monitoring, crop information collection and intelligent path planning. Rai Hari Mohan et al [2] described the current state of agricultural information collection and monitoring, which are still done manually in most areas. Use of IoT and sensors to build a systematic system of work can not only replace this form of work, but also improve the efficiency of the agricultural field. Saha Himadri et al [3], in order to promote smart agriculture and help farmers to cope with natural disasters, developed an intelligent safety monitoring device for agriculture, using an IoT system with corresponding humidity, PH and PIR sensors to establish a complete system to monitor the crop growth process. Wongchai Anupong et al [4] proposed a new technique of integrating soft measurement techniques for remote sensing models based on the intelligent sensing function of AI sensors to improve the accuracy of information collected in agriculture. Ramakrishnam Raju S. V. S et al [5] studied how to develop hydroponic agriculture using IoT and sensor technology. Firstly, real-time sensors, NPK soil, sunlight, turbidity, pH, temperature and camera modules were installed. Secondly deep learning related knowledge is utilized for crop water level health monitoring. Finally farmers can continuously monitor the field status through mobile device applications. Li Xiaofen [6] investigated the development model of smart agriculture to detect national pastoral food security and alleviate the pressure of ensuring food security by packaging the collected data for transmission and acceptance control by means of IoT network connection and sensor collection. The advance of the Internet of Things in smart agriculture has been thoroughly explored by many scholars. The rational use of agronomic monitoring equipment under IoT systems and the orderly sowing, management and cultivation in production can improve the refinement of agricultural production and reduce the economic losses of crops due to environmental, disaster and other adverse factors [7]. Nowadays, under the development trend of IoT smart agriculture, more and more agricultural intelligent mobile robots are implemented to monitor and collect crop information in the field.Song Yun Yun et al [8]. address the problem that mobile robots cannot move effectively in unknown environments, and combine Canny and Otsu to extract obstacle features and find the key pixel locations of obstacles by monocular vision. Then the image depth estimation algorithm is used to estimate the gaps. And an improved bug algorithm is proposed to avoid the obstacle strategy autonomously. Milan SÁGA et al [9]. Considering the simulation of non-stationary random vibration of vertical vehicles with various speeds, the evolved non-stationary random function will be simulated by the variable speed of the vehicle model and the vertical irregularity of the track, the stress and strain analysis of fatigue specimens in the region of the tested gauge section, and the path planning optimization of a six-degree-of-freedom robot manipulator using the evolutionary algorithm. Kameyama

Kentaro et al [10]. proposed a mechanical leg structure to overcome the problem of disabling small robots due to sudden increase in water depth in the field, which allows the wheels to reach the ground by deforming the legs that support them (variable legs). After comparison. It was verified that the variable-leg model could operate in 180 mm water depth, while the fixed-leg model could only operate in about 80 mm water depth. From the design point of view, the application of IoT in smart agriculture should not be limited to the optimization of technology as well as algorithms, but also needs to pay attention to other issues that will be encountered in the use of the product, such as ease of use, user satisfaction, design integrity and other issues. In addition, farmers are less able to organize and analyze complex data due to their limited literacy level, and at the same time, precise operational requirements can discourage their use. At present, there is less literature on obtaining evaluation indicators based on the characteristics of farmers as a user group and making preferential decisions on design solutions based on the ranking of evaluation indicator weight. This leads to low practicality of the relevant products when they flow into the market, which indirectly affects the long-term development of IoT agricultural monitoring machines.

Product development is an ongoing iterative process from design to implementation to testing. During the development process, designers should be closely related to users, and effectively extracting user requirements is often difficult for designers to accomplish. To alleviate this problem, numerous scholars have collected and delineated user requirements through questionnaires and user interviews, however, in the implementation process, researchers generally pre-designed the response range, making the feedback offered by the respondents limited. At the same time, the researcher subjectively divides the collected demand information into categories, which are likely to lead to an unreasonable division. As a result, the obtained results are often influenced by the personal wishes of the researcher and the sample size, and lack a certain degree of objectivity and scientifically. In 1984, Noriaki Kano first published a paper on the Kano model in Japan, which divided the need affecting user satisfaction into five categories: basic needs, expectation need, charm need, undifferentiated need, and reverse need. Hu Zhanmei [11] used rough set theory and Kano model to classify users' demand information for the user demand problem of jewelry packaging, and proposed a jewelry packaging design method for users' expectation value. Yang Hao et al [12] used perceptual engineering and the Kano model to quantify customer perceptions of demand types to derive a prioritized ranking of UAV styling elements, and compared them with preference score for validation. Wu Xiaoli et al [13] integrated three methods. Kano, QFD and FAST, to construct design models oriented to user requirements and applied them to the design of smart baby carriages. Bing Yuan et al [14] combined Kano model and AHP method to study the styling design of agricultural machinery in depth from user requirements. H. C. Yadav et al [15] developed a framework for customer satisfaction corresponding to design requirements based on the Kano model, which was used to explore the aesthetic design of the car from and illustrated with examples. Xian Wang et al [16] derived user preferences based on the Kano model and improved the design of outdoor water purifiers accordingly. Combined with the above-mentioned scholars' research, it can be seen that the Kano model can effectively classify user needs categories in practical application, which is conducive to researchers to dig out implicit feedback such as users' desired needs and preferences in the design stage, so as to provide a decision basis for developing and designing new products.

In the early stage of product development, on the one hand, designers need in order to achieve precise positioning and delineation of customer needs in order to propose multiple design solutions in conjunction with reality. On the other hand, finding the best design solution that satisfies all the design constraints is quite an important step, and the reasonable decision result serves to promote the science of the product and also influences the future iterative

design direction of the product. For complex design problems, designers can use multiple decision methods to achieve evaluation and solution preference. The decision methods commonly used to determine the preferred design solution are: AHP method [17, 18], entropy weighting [19, 20], approximate ideal solution ranking [21, 22], multi-criteria compromise solution ranking [23, 24], and gray correlation analysis [25]. In addition, in order to prevent overly subjective assignment results in the comprehensive evaluation, many scholars use the combined assignment method and apply this method in research experiments. Liang Shiyuan et al [26] evaluated the ship engine simulator based on the combined assignment to improve the rationality of the related index assignment. Hu Yu shan et al [27] established a credit evaluation model for road transport enterprises based on the combined assignment method, which fully considered the degree of differences between indicators and compensated for the possible bias of a single assignment method. Combined the fuzzy AHP method and the improved entropy weight method to calculate the weight values of each index in the comprehensive assessment of power quality of DC power supply. Yin Hodong et al [28] integrated subjective empowerment-AHP and objective empowerment-entropy weight method to establish an EDM machine tool design solution evaluation index system from five levels, including physical scale, safety, function and operation, aesthetics, and operator expectation, and performed comprehensive index weight calculation. The above-mentioned studies were performed by combining the index weight values derived from the subjective assignment method and the objective assignment method to achieve a more accurate assignment. The method can compensate the defects of each of the two types of methods in the process of empowerment, so as to obtain reasonable decision consequences.

The VIKOR method is a multi-attribute optimization decision method that is commonly used in several disciplines. The specific steps are: comparison by ranking the group utility value, regret value and combined utility value of each solution, and ranking the superiority and inferiority of the solutions built on the comparison results. Abhishek Guleria et al [29] combined the VIKOR and TOPSIS methods to make an appropriate assessment of the suitability of a site as a hydrogen power plant site based on some basic criteria, expert opinion of decision makers and other qualitative or quantitative factors. R. Rajesh [30] proposed a model combining a gray clustering algorithm and the VIKOR method for classifying and evaluating barrier problems for supply chain resilience. Grey clustering algorithm is first used to initially rank the barrier problems, and then the VIKOR method is applied to the selected barrier problems to prioritize them efficiently. Wang Zhiyuan et al [31] proposed the idea of design solution evaluation and preference combining entropy power and VIKOR method, and carried out validation calculations with the design solution evaluation of home purification and disinfection as an example, and the results proved that VIKOR method can improve the objectivity and accuracy of solution evaluation results. Tsung-Han Chang [32] evaluated the quality of hospital services based on the evaluation framework of fuzzy set theory and the VIKOR method. Gao Pei [33] performed intuitive fuzzy multi-attribute group decision making based on the VIKOR method and applied it to the evaluation of the quality of university English teaching. Combined with the above literature, it is clear that the VIKOR method maximizes the group benefits and minimizes the individual regret of the opposing views, resulting in an acceptable compromise solution with priority ranking. Therefore, it can obtain the preferred solution closest to the ideal solution of the decision problem of the scheme, with some ranking stability and superiority.

In summary, we choose the combined assignment method and VIKOR method for comprehensive evaluation and decision making of agricultural intelligent robot solutions. First, the Kano model is used to obtain and classify user requirements, and the classified requirements are utilized to establish a solution requirement system. Then, we use the combination

assignment method to assign scientific weight to each indicator, and designers produce multiple solutions according to the importance of each requirement indicator. Finally, the VIKOR method is implemented to obtain the group utility values and rank the solutions to select the most desirable one. Although the VIKOR method has some superiority in decision making, it continues to have some drawbacks when used alone. Therefore, a decision model combining the combined assignment method and the VIKOR method is therefore proposed to remedy the problems that the VIKOR method would have when used alone. The method improves user satisfaction and the scientifically of decision making in solution design, and provides a theoretical reference for designers to study the design requirements of agricultural intelligent robots, which can make the designed products better serve farmers as a group. Also, it helps designers to discover useful design patterns and models, which are essential for developing new design methods.

The main contributions of this study include the following.

1. In the context of the Internet of Things, the agricultural production process presents intelligence, we analyze the user needs of artificial intelligence products in smart agriculture, and discuss in detail the evaluation process of agricultural intelligent robot design integrating Kano model, combined empowerment method and VIKOR method

2. First, a comprehensive evaluation index system is established, and the Kano model is used to divide the user requirements from the main three requirement attributes. The hierarchy of indicators is systematically divided.

3. Second, we combine and subjective empowerment with objective empowerment method to make the empowerment results more scientific.

4. Finally, based on the importance ranking of demand indicators, we designed several solutions, and combined the assignment results with the VIKOR method to select the best solution and refine the design of the solution products.

## 2 Research process

The research process of design solution evaluation of agricultural intelligent robots is divided into three main parts:

1. Dividing user demand attributes based on the Kano model and constructing the evaluation index system of agricultural intelligent gathering robot solutions.

2. The combination of subjective assignment (AHP method) and objective assignment (entropy method) is used to calculate the comprehensive weight of each index, and multiple design solutions are output based on the ranking of the comprehensive weight.

3. VIKOR is applied to find out the group utility value, regret value and comprehensive utility value of each scheme, and carry out comprehensive ranking. The specific design solution flow is shown in Fig 1.

## 3 User requirements analysis based on Kano model

### 3.1 User needs analysis

Due to the advantages of flexibility and convenience of small robots in outdoor fields. They are commonly used in existing agricultural smart robots for monitoring the growth of crops. Through user interviews and market research and other methods, we summarize and analyze

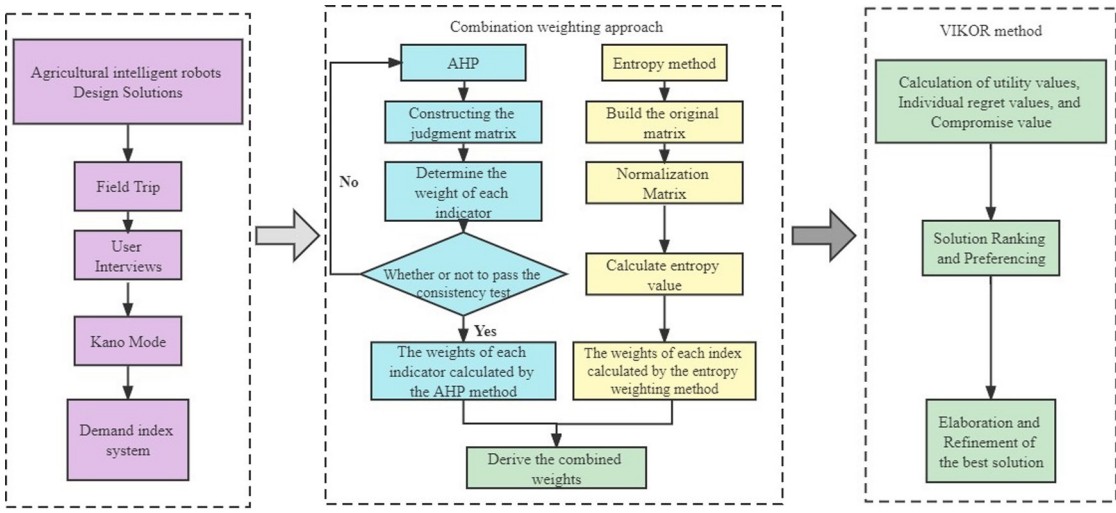

**Fig 1. Research flow chart.**

the agricultural intelligent collection robots of several companies to derive user satisfaction points and expectation points. The target user population was set at agricultural workers, and a total of 260 questionnaires were distributed to the relevant user population during field visits to anticipate the possible user needs. By eliminating user needs that are not related to the agricultural intelligent robots, the questionnaire results yielded a total of 22 initial need, which was classified and processed to obtain four attribute aspects, as showed in Table 1.

## 3.2 Categorizing requirement attributes using the Kano model

The Kano model [34, 35] is a method for classifying and prioritizing user requirements, enabling qualitative and quantitative requirements analysis, and is now widely used in determining product requirements. The Kano model questionnaire was set up based on the content of the initial user requirements form, and each question in the questionnaire had one positive and one negative classification, and the user selected each question based on five rating criteria (satisfied, desirable, no impact, acceptable, unsatisfactory). The questionnaire is able to reflect the user's satisfaction that the agricultural intelligent robot has a certain need or does not have a certain need, corresponding to the form of the questionnaire, as showed in Fig 2. The research study was mainly conducted online by inviting users to fill out 400 questionnaires, excluding invalid results questionnaires, the total number of valid questionnaires collected 378, 68% of the questionnaires were returned by men, 32% by women, and the age group was distributed between 20–65 years old. The questionnaire results of each demand are categorized

**Table 1. User initial requirements form.**

| Security | Functionality | Aestheticity | Interactivity |
|---|---|---|---|
| Drive steady | Data Processing | Fluent modeling | Voice Interaction |
| Intelligent avoidance | Identifying pests | Reasonable structure | Remote operation |
| Non-destructive | Timed charging | Durable material | Preset work |
| Facial Recognition | Automatic navigation | Color coordination | Button layout |
| Anti-theft system | Information Collection | Warning Appearance | Image Interaction |
| Terrain adaptation | Shared use | / | / |

| Demand issues | Satisfied | As it should be | No effect | Can accept | Dissatisfied |
|---|---|---|---|---|---|
| Provides some kind of demand | | ✓ | | | |
| Does not provides some kind of demand | | | | ✓ | |

**Fig 2. Kano model questionnaire.**

according to the Kano model evaluation criteria, and the satisfaction attribute of the demand with the most selected attributes among all the results of the questionnaire is divided into satisfaction attributes M, O, A, I, R, Q, in order (M is a must-have demand; O is; expectation demand; A is a charm demand; I is a non-differentiated demand; R is a reverse demand; Q is a doubtful result) to divide the evaluation criteria as shown in Fig 3.

The statistical results after sorting and categorizing, the detailed data are shown in Table 2. Where the basic-type attributes (M) include drive steady, data processing, identifying pests, information collection and remote operation. Desired attributes (O) include image interaction, intelligent avoidance, terrain adaptation, reasonable structure, button layout and automatic navigation. Charismatic attributes (A) include fluent modeling, color coordination, anti-theft system, preset work and durable material. In the Kano model, basic attributes are usually among the first design points that designers consider. Desired attributes are not required for design, but if satisfied will significantly increase user satisfaction. Attractive attributes are attributes that users do not particularly expect, but if they are satisfied, they can bring a surprise to users, and then user satisfaction will be increased accordingly. Their importance is M, O and A in order, and the corresponding undifferentiated attributes (I) and reversed attributes (R) can be disregarded in the design of this paper.

| | Demand issues | Provides some kind of demand | | | | |
|---|---|---|---|---|---|---|
| | | Satisfied | As it should be | No effect | Can accept | Dissatisfied |
| Does not provides some kind of demand | Satisfied | Q | A | A | A | O |
| | As it should be | R | I | I | I | M |
| | No effect | R | I | I | I | M |
| | Can accept | R | I | I | I | M |
| | Dissatisfied | R | R | R | R | Q |

**Fig 3. Kano model evaluation criteria.**

**Table 2. Summary of user satisfaction based on Kano model.**

| Serial number | Classification | Demand | M | O | A | I | R | Q | Demand Properties |
|---|---|---|---|---|---|---|---|---|---|
| 1 | Security | Drive steady | 246 | 45 | 31 | 24 | 19 | 13 | M |
| 2 | | Intelligent avoidance | 92 | 157 | 53 | 46 | 21 | 9 | O |
| 3 | | Non-destructive | 87 | 74 | 58 | 132 | 15 | 12 | I |
| 4 | | Facial Recognition | 64 | 82 | 40 | 182 | 3 | 7 | I |
| 5 | | Anti-theft system | 51 | 67 | 194 | 43 | 12 | 11 | A |
| 6 | | Terrain adaptation | 82 | 183 | 44 | 56 | 8 | 5 | O |
| 7 | Functionality | Data Processing | 206 | 56 | 47 | 49 | 13 | 7 | M |
| 8 | | Identifying pests | 269 | 40 | 24 | 21 | 18 | 6 | M |
| 9 | | Timed charging | 68 | 48 | 33 | 203 | 16 | 10 | I |
| 10 | | Automatic navigation | 78 | 174 | 83 | 29 | 9 | 5 | O |
| 11 | | Information Collection | 251 | 51 | 32 | 31 | 7 | 6 | M |
| 12 | | Shared use | 47 | 64 | 71 | 184 | 5 | 7 | I |
| 13 | Aestheticity | Fluent modeling | 94 | 72 | 132 | 51 | 19 | 10 | A |
| 14 | | Reasonable structure | 73 | 164 | 52 | 72 | 8 | 9 | O |
| 15 | | Durable material | 72 | 93 | 115 | 88 | 6 | 4 | A |
| 16 | | Color coordination | 57 | 101 | 163 | 43 | 3 | 11 | A |
| 17 | | Warning Appearance | 65 | 47 | 54 | 198 | 9 | 5 | I |
| 18 | Interactivity | Voice Interaction | 28 | 78 | 61 | 186 | 14 | 11 | I |
| 19 | | Remote operation | 227 | 56 | 49 | 26 | 12 | 8 | M |
| 20 | | Preset work | 77 | 82 | 149 | 55 | 9 | 6 | A |
| 21 | | Button layout | 45 | 154 | 97 | 72 | 6 | 4 | O |
| 22 | | Image Interaction | 69 | 171 | 95 | 34 | 4 | 5 | O |

## 3.3 Construct the evaluation system of agricultural intelligent robots demand index

The evaluation and preference of product design solutions should consider multiple factors based on user need and build the index system from the three principles of refinement and comprehensiveness operability [36]. Considering that each of the demand attributes of the fixed agricultural intelligent robot aggregated by the Kano model contains multiple sub-requirements, here the index system of categorized requirements is established using AHP method to further prioritize the user requirements. The basic type attributes, expectation-type attributes, and charm-type attributes with large influence factors are selected as the criterion layers of the evaluation system, and the corresponding requirements of each criterion layer are sub-criteria layers totaling 16.Indicators under the basic type attribute (M) after numbering are: drive steady ($H_1$), data processing ($H_2$), identifying pests ($H_3$), information collection ($H_4$) and remote operation ($H_5$). The indicators under the expectation-based attribute (O) are image interaction ($H_6$), intelligent avoidance ($H_7$), terrain adaptation ($H_8$), reasonable structure ($H_9$), button layout ($H_{10}$) and automatic navigation ($H_{11}$). The indicators under the charismatic attribute (A) are fluent modeling ($H_{12}$), color coordination ($H_{13}$), anti-theft system ($H_{14}$), preset work ($H_{15}$) and durable material ($H_{16}$). The specific evaluation system, as shown in Fig 4.

## 4 Methodology for evaluating intelligent robotic solutions in agriculture

### 4.1 Subjective weighting—AHP method to determine the weight of each index

The core of the AHP method is hierarchical weighted decision analysis, which constructs a hierarchical structure model of the elements related to decision making as a basis for

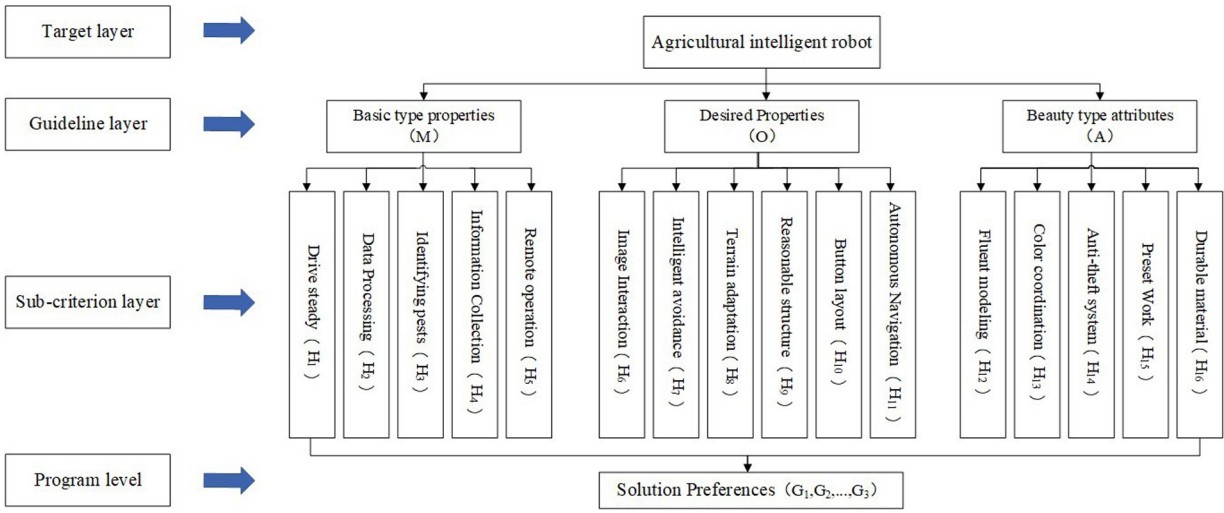

**Fig 4. Design evaluation index system chart.**

quantitative analysis. The specific steps are: establishing an evaluation system, constructing a two-by-two judgment matrix, calculating weights, testing consistency, single-level ranking of weights and total ranking. The AHP method, as a commonly used subjective assignment method, has been used by many scholars to calculate the weights of indicators in recent years, and the relevant arithmetic steps and formulas of the AHP method in the article are referenced in the literature [37, 38].

## 4.2 Objective weighting—Entropy weighting method to determine the weight of each index

Entropy weighting method is the weight coefficient of each index determined by the information entropy value, and the calculation process does not introduce subjective assumptions, so the obtained index weighting value has a certain objectivity. The specific steps are: inviting experts to score, constructing the scoring matrix, normalizing the scoring matrix data, solving the entropy value of each indicator, and calculating the weight of each indicator. The relevant operational steps and formulas of the entropy weight method in the article are referred to the literature [39, 40].

## 4.3 Determine the combined weight value

The AHP method effectively reflects the intention of experts and users in the process of empowerment, but is relatively subjective. While the entropy value method uses the calculated size of the information entropy of each index in the process of assigning weights, the process is more objective, but to a certain extent ignores the subjective intention of the designer. To address the limitations of using a single method for assigning weights, a combination of subjective (AHP) and objective (entropy method) assignments is used, integrating expert experience with objective mathematical theoretical basis, which can make the weights more accurate and more in line with practical needs. The combined weight $W_j$ is obtained by linearly synthesizing the weight $W_{cj}$ obtained by the AHP method and the weight $W_{dj}$ obtained by the entropy weight method through the formula.

$$W_j = \beta W_{cj} + (1 - \beta) W_{dj} \tag{1}$$

Where: $W_j$ is the combined weight of the $j$th indicator; $\beta$ is the decision preference coefficient ($0 \leq \beta \leq 1$).

## 4.4 VIKOR method to determine the ideal solution

The calculation of the decision making using the VIKOR method proceeds as follows.

(1) Establish a standardized decision matrix.

Assuming that evaluators assess $n$ evaluation indicators for $m$ programs, create the original matrix.

$$V = \left(v_{ij}\right)_{m \times n} = \begin{bmatrix} v_{11} & v_{12} & \cdots & v_{1n} \\ v_{21} & v_{22} & \cdots & v_{2n} \\ \vdots & \vdots & \ddots & \vdots \\ v_{m1} & v_{m2} & \cdots & v_{mn} \end{bmatrix} \tag{2}$$

Where: $v_{ij}$ is the group decision value of the $j$th evaluation indicator of option $v_i$ for all decision makers.

Based on Eqs (4) and (5), the benefit-based and cost-based matrices are normalized to obtain the final matrix $Y$.

$$Y = \left(y_{ij}\right)_{m \times n} = \begin{bmatrix} y_{11} & y_{12} & \cdots & y_{1n} \\ y_{21} & y_{22} & \cdots & y_{2n} \\ \vdots & \vdots & \ddots & \vdots \\ y_{m1} & y_{m2} & \cdots & y_{mn} \end{bmatrix} \tag{3}$$

$$y_{ij} = \frac{v_{ij}}{\sqrt{\sum_{i=1}^{m} \left(v_{ij}\right)^2}} \tag{4}$$

$$y_{ij} = \frac{\frac{1}{v_{ij}}}{\sqrt{\sum_{i=1}^{m} \left(\frac{1}{v_{ij}}\right)^2}} \tag{5}$$

(2) Determine the positive ideal solution $y^+$ and the negative ideal solution $y^-$.

$$\begin{cases} y_j^+ = \max y_{ij} \\ y_j^- = \min y_{ij} \end{cases} \tag{6}$$

(3) Calculate the utility value $S_i$, individual regret value $R_i$ and trade-off value $Q_i$ of the

alternative.

$$S_i = \sum_{j=1}^{n} W_j \frac{y_i^+ - y_{ij}}{y_i^+ - y_i^-} \tag{7}$$

$$R_i = \max\left( W_j \frac{y_i^+ - y_{ij}}{y_i^+ - y_i^-} \right) \tag{8}$$

$$Q_i = \varepsilon \frac{S_i - S^-}{S^+ - S^-} + (1 - \varepsilon)\frac{R_i - R^-}{R^+ - R^-} \tag{9}$$

Where: $S^+ = \max_i \{S_i\}$, $S^- = \min_i \{S_i\}$, $R^+ = \max_i \{R_i\}$, $R^- = \min_i \{R_i\}$, $\varepsilon$ denotes the trade-off coefficient, which is generally taken as $\varepsilon = 0.5$ when making decisions.

(4) Determine the optimal solution.

First, the alternatives are ranked in decreasing order by the results of $S_i$, $R_i$, and $Q_i$ calculations. Second, the options are ranked according to their $Q_i$ values from smallest to largest, respectively: $G_1, G_2, \ldots, G_m$. $m$ is the number of options. Finally, the optimal solution needs to satisfy two conditions, condition 1: $Q(G_1) - Q(G_2) \geq \frac{1}{m-1}$; and condition 2: the stability of the solution is acceptable, the solution $G_1$ that satisfies the minimum value of $Q_i$ also satisfies that one of the values in $S_i$ or $R_i$ is also minimum, if both of the above conditions are met, the best solution is the best solution. If either of the conditions does not match, a compromise set is proposed. If condition 1 is not met, then $\{G_1, G_2, \ldots, G_m\}$ determines the maximum value of $m$ by $Q(z_1) - Q(z_2) \geq \frac{1}{m-1}$, so that the compromise solution can be determined. If condition 2 is not satisfied, then the optimal solution is the solution ranked first and second, then the set of compromise solutions contains $G_1$ and $G_2$.

## 5 Design applications

### 5.1 Calculation of subjective weight values

According to the research and discussion of the designers, it was decided that a total of 5 agricultural workers, 5 agricultural machinery designers, and 5 industrial designers were selected to form a group of 15 scorers. The 15 scorers were given scores by the 1–9 scale method, and the high and low values of the indicators corresponded to their relative importance, and the specific scoring criteria are shown in Table 3. Scoring corresponds to the results of the evaluation matrix of each influential element, as showed in Tables 4–7.

Based on what is described in part 4.1, the evaluation result matrix is first tested for consistency, and the *RI* values corresponding to the judgment matrix are shown in Table 8. If $CR \leq$ 0.1, it indicates that it passes the consistency test, and the test results are shown in Table 9. The results show that all judgment matrices pass the test, and the calculation of the weights of each

**Table 3. 1–9 scoring criteria.**

| Relative Importance Assignment | Scale description (comparison of two factors) |
|:---:|:---:|
| 1 | Equally as important |
| 3 | The former is slightly more important than the latter |
| 5 | The former is significantly more important than the latter |
| 7 | The former is more strongly important than the latter |
| 9 | The former is definitely more important than the latter |
| 2,4,6,8 | The middle value between the two relative |

**Table 4. Target layer judgment matrix and weight values.**

| D | Basic type properties (M) | Desired Properties (O) | Charismatic Properties (A) | Weighting value |
|---|---|---|---|---|
| Basic type properties (M) | 1 | 2 | 3 | 0.5278 |
| Desired Properties (O) | 1/2 | 1 | 3 | 0.3325 |
| Charismatic Properties (A) | 1/3 | 1/3 | 1 | 0.1396 |

**Table 5. Basic type attribute M judgment matrix and weight values.**

| M | Drive steady (H1) | Data Processing (H2) | Identifying pests (H3) | Information Collection (H4) | Remote operation (H5) | Weighting value |
|---|---|---|---|---|---|---|
| Drive steady (H1) | 1 | 1/5 | 1/3 | 1/6 | 1/4 | 0.0471 |
| Data Processing (H2) | 5 | 1 | 4 | 1/2 | 2 | 0.2716 |
| Identifying pests (H3) | 3 | ¼ | 1 | 1/5 | 1/3 | 0.0845 |
| Information Collection (H4) | 6 | 2 | 5 | 1 | 3 | 0.4241 |
| Remote operation (H5) | 4 | ½ | 3 | 1/3 | 1 | 0.1727 |

**Table 6. Expected attribute O judgment matrix and weight values.**

| O | Image Interaction (H6) | Intelligent avoidance (H7) | Terrain adaptation (H8) | Reasonable structure (H9) | Button layout (H10) | Automatic navigation (H11) | Weighting value |
|---|---|---|---|---|---|---|---|
| Image Interaction (H6) | 1 | 1/4 | 1/5 | 1/7 | 1/2 | 1/4 | 0.0380 |
| Intelligent avoidance (H7) | 4 | 1 | 1/2 | 1/4 | 3 | 2 | 0.1448 |
| Terrain adaptation (H8) | 5 | 2 | 1 | 1/2 | 4 | 3 | 0.2363 |
| Reasonable structure (H9) | 7 | 4 | 2 | 1 | 6 | 5 | 0.4192 |
| Button layout (H10) | 2 | 1/3 | 1/4 | 1/6 | 1 | 1/3 | 0.0554 |
| Automatic navigation (H11) | 4 | 1/2 | 1/3 | 1/5 | 3 | 1 | 0.1063 |

**Table 7. Charm attribute A judgment matrix and weight values.**

| A | Fluent modeling (H12) | Color coordination (H13) | Anti-theft system (H14) | Preset work (H15) | Durable material (H16) | Weighting value |
|---|---|---|---|---|---|---|
| Fluent modeling (H12) | 1 | 6 | 4 | 2 | 5 | 0.4364 |
| Color coordination (H13) | 1/6 | 1 | 1/3 | 1/7 | 1/2 | 0.0479 |
| Anti-theft system (H14) | 1/4 | 3 | 1 | 1/3 | 2 | 0.1239 |
| Preset work (H15) | 1/2 | 7 | 3 | 1 | 5 | 0.3167 |
| Durable material (H16) | 1/5 | 2 | 1/2 | 1/5 | 1 | 0.0751 |

**Table 8. Corresponding RI values.**

| n | 3 | 4 | 5 | 6 | 7 | 8 | 9 |
|---|---|---|---|---|---|---|---|
| RI | 0.52 | 0.89 | 1.12 | 1.26 | 1.36 | 1.41 | 1.46 |

Table 9. Consistency test results.

| | G | M | A | O |
|---|---|---|---|---|
| $\lambda_{max}$ | 3.0536 | 5.1644 | 6.2048 | 5.1066 |
| CI | 0.0268 | 0.0411 | 0.0410 | 0.0267 |
| CR | 0.0516 | 0.0367 | 0.0325 | 0.0238 |

index can be continued. The specific results of the subjective weight calculation are shown in the schedule S1 Table in S1 File. The results of the subjective weight calculation and the comparison chart of the weight values of each index are shown in Fig 5.

## 5.2 Calculation of objective weight values

Based on what is described in part 4.2, first, the indicators were scored by 15 decision makers on a percentage scale for 16 indicators, resulting in scoring data to form the $X$. Then, the data are normalized to obtain the matrix $X^*$ Finally, the weights of each evaluation index were calculated. The initial matrix data are shown in Table 10, and the normalized data are shown in Table 11. The calculation was performed using the normalized matrix, and the results of the calculation and comparison of the weight values of each indicator are shown in Fig 6, and the detailed information entropy values and weight values are shown in schedule S2 Table in S1 File.

## 5.3 Determining the combined weights

The proportion of the weights derived from the hierarchical analysis and entropy weighting are measured and calculated based on the combination of Eq (1). Usually, scholars take the preference coefficient $\beta$ to be close to 0.3 in order to increase the share of subjective weights in the composite weights. Conversely, to increase the share of objective weights in the composite weight, β is taken to be close to 0.7. If $\beta$ is 0.5, the proportion of subjective weights and objective weights in the composite weight value is the same. The determination of the AHP risk judgment matrix relies on the experience of experts, which may cause confusion in the judgment of experts due to the large number of evaluation indicators and the large

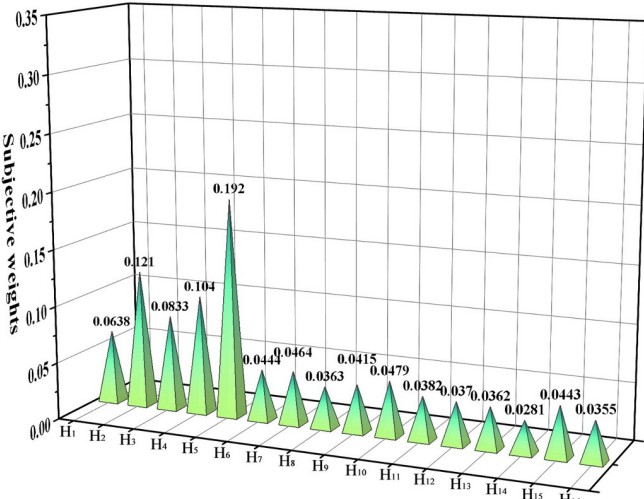

Fig 5. Comparison chart of the weight values of each indicator for subjective weight calculation.

**Table 10. Initial matrix data.**

|  | $H_1$ | $H_2$ | $H_3$ | $H_4$ | $H_5$ | $H_6$ | $H_7$ | $H_8$ | $H_9$ | $H_{10}$ | $H_{11}$ | $H_{12}$ | $H_{13}$ | $H_{14}$ | $H_{15}$ | $H_{16}$ |
|---|---|---|---|---|---|---|---|---|---|---|---|---|---|---|---|---|
| 1 | 80 | 80 | 80 | 85 | 75 | 80 | 70 | 85 | 75 | 75 | 85 | 80 | 80 | 85 | 85 | 85 |
| 2 | 81 | 91 | 80 | 94 | 70 | 87 | 71 | 94 | 93 | 85 | 75 | 87 | 88 | 90 | 92 | 95 |
| 3 | 70 | 80 | 70 | 80 | 80 | 80 | 80 | 90 | 90 | 70 | 80 | 90 | 90 | 80 | 80 | 90 |
| 4 | 80 | 80 | 80 | 80 | 85 | 80 | 60 | 78 | 78 | 65 | 60 | 70 | 60 | 75 | 90 | 88 |
| 5 | 79 | 92 | 83 | 89 | 70 | 86 | 69 | 96 | 91 | 79 | 80 | 90 | 81 | 93 | 93 | 93 |
| 6 | 87 | 89 | 72 | 92 | 72 | 85 | 68 | 89 | 84 | 81 | 72 | 82 | 80 | 88 | 79 | 91 |
| 7 | 79 | 89 | 79 | 96 | 71 | 81 | 86 | 83 | 88 | 82 | 70 | 86 | 72 | 84 | 82 | 89 |
| 8 | 90 | 92 | 79 | 93 | 70 | 83 | 75 | 86 | 94 | 93 | 82 | 89 | 82 | 88 | 86 | 93 |
| 9 | 89 | 94 | 90 | 87 | 71 | 83 | 70 | 86 | 96 | 89 | 75 | 95 | 80 | 91 | 88 | 93 |
| 10 | 79 | 89 | 78 | 95 | 71 | 91 | 75 | 84 | 88 | 78 | 76 | 86 | 82 | 85 | 90 | 86 |
| 11 | 70 | 80 | 80 | 80 | 80 | 80 | 70 | 70 | 70 | 60 | 80 | 90 | 70 | 60 | 70 | 80 |
| 12 | 90 | 91 | 76 | 95 | 86 | 90 | 82 | 93 | 92 | 91 | 88 | 91 | 90 | 89 | 92 | 92 |
| 13 | 79 | 93 | 88 | 89 | 71 | 92 | 71 | 91 | 91 | 94 | 89 | 88 | 86 | 89 | 91 | 88 |
| 14 | 85 | 82 | 75 | 89 | 76 | 78 | 69 | 83 | 89 | 82 | 71 | 76 | 74 | 88 | 76 | 91 |
| 15 | 84 | 90 | 70 | 82 | 81 | 72 | 71 | 83 | 87 | 82 | 80 | 86 | 82 | 82 | 79 | 93 |

scalar workload for the design of agricultural intelligent robots. And the entropy weight method has certain requirements for the discrete degree of the relevant index values, and for indicators with small changes, the objective weight calculation results may not necessarily meet the reality, so, in order to maximize the retention of the respective information of the two weight values and give full play to their respective advantages, it is finally determined that β is taken as 0.5 in this paper in combination with reference [41]. The results of the combined weight calculation are shown in Table 12, and most of the combined weights of indicators obtained by the combined weighting method are between the subjective and objective weights, indicating that the combined weights take into account the importance expression of subjective and objective weights. The comparison of subjective weights, objective weights and combined weights is shown in Fig 7.

**Table 11. Matrix data after normalization.**

|  | $H_1$ | $H_2$ | $H_3$ | $H_4$ | $H_5$ | $H_6$ | $H_7$ | $H_8$ | $H_9$ | $H_{10}$ | $H_{11}$ | $H_{12}$ | $H_{13}$ | $H_{14}$ | $H_{15}$ | $H_{16}$ |
|---|---|---|---|---|---|---|---|---|---|---|---|---|---|---|---|---|
| 1 | 0.50 | 0.00 | 0.50 | 0.31 | 0.31 | 0.40 | 0.38 | 0.58 | 0.19 | 0.44 | 0.86 | 0.40 | 0.67 | 0.76 | 0.65 | 0.33 |
| 2 | 0.55 | 0.79 | 0.50 | 0.88 | 0.00 | 0.75 | 0.42 | 0.92 | 0.88 | 0.74 | 0.52 | 0.68 | 0.93 | 0.91 | 0.96 | 1.00 |
| 3 | 0.00 | 0.00 | 0.00 | 0.00 | 0.63 | 0.40 | 0.77 | 0.77 | 0.77 | 0.29 | 0.69 | 0.80 | 1.00 | 0.61 | 0.43 | 0.67 |
| 4 | 0.50 | 0.00 | 0.50 | 0.00 | 0.94 | 0.40 | 0.00 | 0.31 | 0.31 | 0.15 | 0.00 | 0.00 | 0.00 | 0.45 | 0.87 | 0.53 |
| 5 | 0.45 | 0.86 | 0.65 | 0.56 | 0.00 | 0.70 | 0.35 | 1.00 | 0.81 | 0.56 | 0.69 | 0.80 | 0.70 | 1.00 | 1.00 | 0.87 |
| 6 | 0.85 | 0.64 | 0.10 | 0.75 | 0.13 | 0.65 | 0.31 | 0.73 | 0.54 | 0.62 | 0.41 | 0.48 | 0.67 | 0.85 | 0.39 | 0.73 |
| 7 | 0.45 | 0.64 | 0.45 | 1.00 | 0.06 | 0.45 | 1.00 | 0.50 | 0.69 | 0.65 | 0.34 | 0.64 | 0.40 | 0.73 | 0.52 | 0.60 |
| 8 | 1.00 | 0.86 | 0.45 | 0.81 | 0.00 | 0.55 | 0.58 | 0.62 | 0.92 | 0.97 | 0.76 | 0.76 | 0.73 | 0.85 | 0.70 | 0.87 |
| 9 | 0.95 | 1.00 | 1.00 | 0.44 | 0.06 | 0.55 | 0.38 | 0.62 | 1.00 | 0.85 | 0.52 | 1.00 | 0.67 | 0.94 | 0.78 | 0.87 |
| 10 | 0.45 | 0.64 | 0.40 | 0.94 | 0.06 | 0.95 | 0.58 | 0.54 | 0.69 | 0.53 | 0.55 | 0.64 | 0.73 | 0.76 | 0.87 | 0.40 |
| 11 | 0.00 | 0.00 | 0.50 | 0.00 | 0.63 | 0.40 | 0.38 | 0.00 | 0.00 | 0.00 | 0.69 | 0.80 | 0.33 | 0.00 | 0.00 | 0.00 |
| 12 | 1.00 | 0.79 | 0.30 | 0.94 | 1.00 | 0.90 | 0.85 | 0.88 | 0.85 | 0.91 | 0.97 | 0.84 | 1.00 | 0.88 | 0.96 | 0.80 |
| 13 | 0.45 | 0.93 | 0.90 | 0.56 | 0.06 | 1.00 | 0.42 | 0.81 | 0.81 | 1.00 | 1.00 | 0.72 | 0.87 | 0.88 | 0.91 | 0.53 |
| 14 | 0.75 | 0.14 | 0.25 | 0.56 | 0.38 | 0.30 | 0.35 | 0.50 | 0.73 | 0.65 | 0.38 | 0.24 | 0.47 | 0.85 | 0.26 | 0.73 |
| 15 | 0.70 | 0.71 | 0.00 | 0.13 | 0.69 | 0.00 | 0.42 | 0.50 | 0.65 | 0.65 | 0.69 | 0.64 | 0.73 | 0.67 | 0.39 | 0.87 |

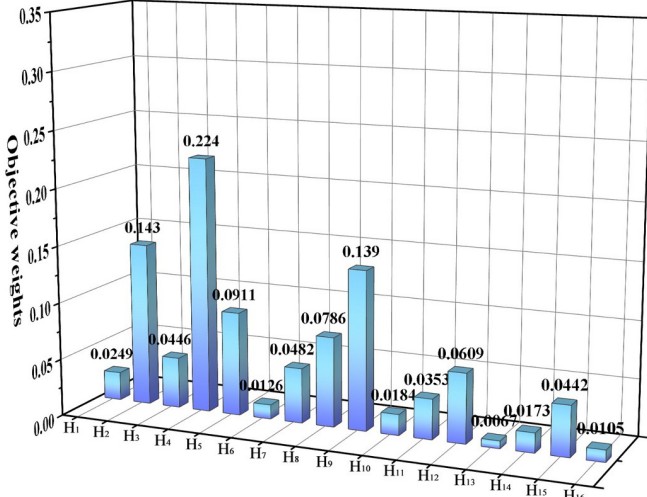

**Fig 6. Comparison chart of weight values of each indicator for objective weighting calculation.**

## 5.4 Exporting multiple design solutions

The weight coefficients of each evaluation index and the ranking of importance were obtained from the above calculation, according to which the order of importance of the evaluation indexes of the agricultural intelligent robots design scheme is: Information collection>Remote operation>Data processing>Rational structure> Identification pests>Terrain adaptation>Fluent modeling>Intelligent avoidance> Drive driving>Preset work>Autonomous navigation>Button layout>Image interaction>Durable material>Anti-theft system>Color coordination. Based on the results of the comprehensive ranking of design elements, the design solution output for the agricultural information collection robot was finally derived from three different design solutions, as shown in Fig 8.

**Table 12. Combined weight calculation results and ranking.**

| Evaluation Indicators | Weighting value (AHP) | Weighting value (Entropy method) | Combined weighting value | Sorting |
|---|---|---|---|---|
| $H_1$ | 0.0249 | 0.0638 | 0.0444 | 9 |
| $H_2$ | 0.1433 | 0.1208 | 0.1321 | 3 |
| $H_3$ | 0.0446 | 0.0833 | 0.0640 | 5 |
| $H_4$ | 0.2239 | 0.1039 | 0.1639 | 1 |
| $H_5$ | 0.0911 | 0.1924 | 0.1418 | 2 |
| $H_6$ | 0.0126 | 0.0444 | 0.0285 | 13 |
| $H_7$ | 0.0482 | 0.0464 | 0.0473 | 8 |
| $H_8$ | 0.0786 | 0.0363 | 0.0575 | 6 |
| $H_9$ | 0.1394 | 0.0415 | 0.0904 | 4 |
| $H_{10}$ | 0.0184 | 0.0479 | 0.0331 | 12 |
| $H_{11}$ | 0.0353 | 0.0382 | 0.0368 | 11 |
| $H_{12}$ | 0.0609 | 0.0370 | 0.0490 | 7 |
| $H_{13}$ | 0.0067 | 0.0362 | 0.0215 | 16 |
| $H_{14}$ | 0.0173 | 0.0281 | 0.0227 | 15 |
| $H_{15}$ | 0.0442 | 0.0443 | 0.0443 | 10 |
| $H_{16}$ | 0.0105 | 0.0355 | 0.0230 | 14 |

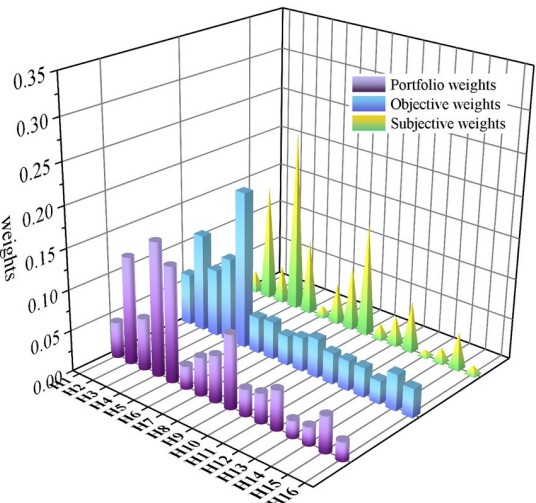

**Fig 7. Comparison chart of the three weight values.**

All three design solutions start from the importance of the design factors of the indicator, solution $G_1$ adopt the wheel type of movement, The robotic arm that collects information about the crop is located right in the middle of the top of the product, The end of the product is equipped with an automatic folding retrievable soil information collection device, The camera and interactive control switch is located at the front of the product. Solution $G_2$ uses a tracked design, The robotic arm is located in the rear half of the product, Soil information collection device placed at the end of the product, the interactive touch screen is located in front of the product and has a rotatable camera; Solution $G_3$ is also of tracked design, however, there is a difference with the track type of Solution $G_2$, the robot arm is located on the left side of the product, a device for collecting soil information of relative size is placed at the end of the product. The camera function is located in the head of the product, the interaction area is the top button and the front touch screen part, respectively.

## 5.5 Program evaluation and preference

The three programs were evaluated by 15 raters for preferential selection, and each design evaluation index was assigned a value using a 9-level scale. The average value of the assignment

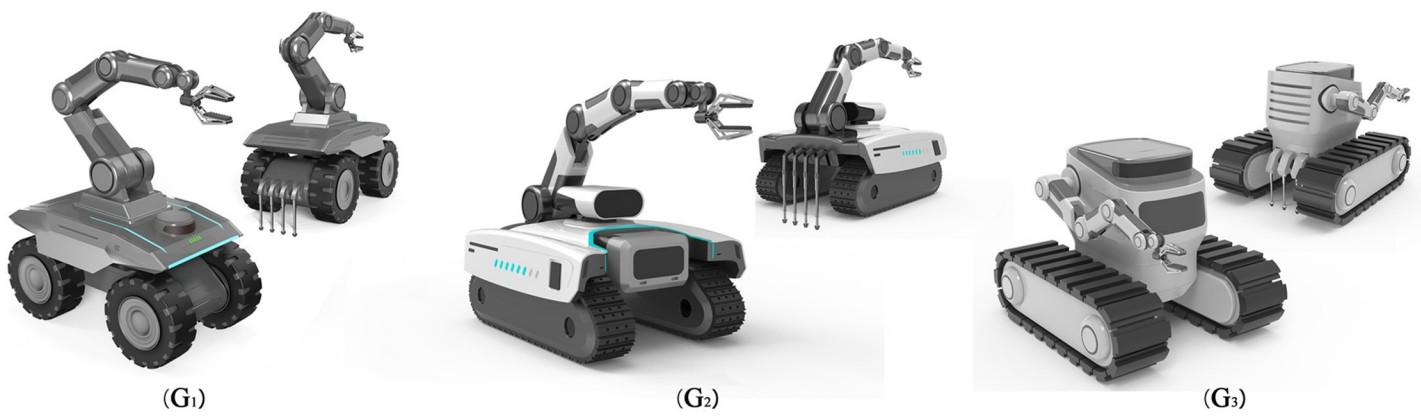

$(G_1)$                                   $(G_2)$                                   $(G_3)$

**Fig 8. Multiple scheme design drawings.**

results was calculated as the initial evaluation value of each program, and the initial evaluation matrix was established.

$$V = \begin{bmatrix} 7.7 & 7.9 & 8.0 & 7.7 & 8.0 & 7.6 & 7.0 & 7.6 & 7.5 & 7.8 & 7.2 & 7.6 & 8.0 & 7.1 & 7.2 & 7.2 \\ 7.9 & 8.1 & 7.9 & 7.8 & 8.2 & 8.0 & 7.3 & 7.8 & 8.1 & 8.3 & 7.4 & 7.8 & 8.3 & 7.0 & 7.6 & 7.5 \\ 7.2 & 6.4 & 7.2 & 7.5 & 8.1 & 7.5 & 7.7 & 7.4 & 7.2 & 7.4 & 6.9 & 6.5 & 7.7 & 7.3 & 7.1 & 7.4 \end{bmatrix}$$

Based on the VIKOR method, all evaluation indicators in the evaluation system are in the form of benefit type, so they are normalized according to Eq (4). The following matrix was obtained.

$$Y = \begin{bmatrix} 0.585 & 0.608 & 0.599 & 0.580 & 0.570 & 0.570 & 0.551 & 0.577 & 0.569 & 0.574 & 0.580 & 0.599 & 0.577 & 0.575 & 0.569 & 0.564 \\ 0.600 & 0.623 & 0.592 & 0.587 & 0.584 & 0.600 & 0.574 & 0.592 & 0.615 & 0.611 & 0.596 & 0.615 & 0.599 & 0.566 & 0.601 & 0.588 \\ 0.547 & 0.492 & 0.539 & 0.565 & 0.577 & 0.562 & 0.606 & 0.562 & 0.546 & 0.545 & 0.556 & 0.513 & 0.555 & 0.591 & 0.561 & 0.580 \end{bmatrix}$$

The group utility value $S_i$, individual regret value $R_i$ and trade-off value $Q_i$ were calculated for each scenario using the normalized matrix combined with the combined weights of each indicator using Eqs (7)–(9), respectively. the closer the optimal solution is, the calculation results show that $G_2 < G_1 < G_3$. Solution $G_2$ satisfies both conditions of the VIKOR optimal solution selection. Condition 1: $Q(G_1) - Q(G_2) \geq \frac{1}{m-1}, m = 3$ condition is met. Condition 2: Solution $G_2$ corresponds to the best ranking of $S_i$, $R_i$, and $Q_i$ values. Therefore, $G_2$ is the best design scheme, and the specific evaluation calculation results, as shown in Table 13.

## 5.6 Validation of evaluation results

To verify the rationality and scientific validity of the evaluation framework based on Kano model, combined assignment method and VIKOR method in the paper, the fuzzy integrated evaluation method is applied here with the help of the concept of fuzzy mathematics. Twenty target users were invited to rate each of the above three design solutions and create a set of comments $\theta$ = {very welcome, welcome, average, somewhat unwelcome, not at all welcome}, and give them a score of 90, 75, 60, 45 and 30 in that order. Taking the scheme $G_1$ matrix as an example, the must-have type attribute evaluation matrix is denoted by $Z_M$ the expectation type attribute evaluation matrix is denoted by $Z_O$, and the charisma type attribute evaluation matrix

**Table 13. Calculated results for each program evaluation.**

| | Group utility value $S_i$ | Individual regret value $R_i$ | Compromise values $Q_i$ | Program prioritization |
|---|---|---|---|---|
| Solution $G_1$ | 0.5086 | 0.1418 | 0.7061 | 2 |
| Solution $G_2$ | 0.0577 | 0.0270 | 0.0000 | 1 |
| Solution $G_3$ | 0.8438 | 0.1639 | 1.0000 | 3 |

is denoted by $Z_A$.

$$Z_M = \begin{bmatrix} 0.65 & 0.1 & 0.15 & 0.1 & 0.00 \\ 0.45 & 0.4 & 0.15 & 0.00 & 0.00 \\ 0.5 & 0.3 & 0.1 & 0.05 & 0.05 \\ 0.55 & 0.2 & 0.2 & 0.05 & 0.00 \\ 0.5 & 0.2 & 0.15 & 0.15 & 0.00 \end{bmatrix} \tag{10}$$

$$Z_O = \begin{bmatrix} 0.45 & 0.1 & 0.1 & 0.25 & 0.1 \\ 0.65 & 0.25 & 0.05 & 0.05 & 0.00 \\ 0.7 & 0.1 & 0.15 & 0.05 & 0.00 \\ 0.35 & 0.25 & 0.3 & 0.1 & 0.00 \\ 0.4 & 0.2 & 0.15 & 0.2 & 0.05 \\ 0.45 & 0.25 & 0.25 & 0.05 & 0.00 \end{bmatrix} \tag{11}$$

$$Z_A = \begin{bmatrix} 0.65 & 0.2 & 0.1 & 0.05 & 0.00 \\ 0.35 & 0.35 & 0.25 & 0.05 & 0.00 \\ 0.45 & 0.25 & 0.2 & 0.1 & 0.00 \\ 0.7 & 0.15 & 0.1 & 0.05 & 0.00 \\ 0.35 & 0.45 & 0.15 & 0.05 & 0.00 \end{bmatrix} \tag{12}$$

The weights and evaluation matrices of each of the above indicators are brought into the following formula to calculate the secondary evaluation model vectors $P_M$, $P_O$ and $P_A$, respectively, and the results are as follows.

$$P_M = W_M \times R_M = (0.530,\ 0.240,\ 0.150,\ 0.070,\ 0.010) \tag{13}$$

$$P_O = W_O \times R_O = (0.495,\ 0.200,\ 0.188,\ 0.102,\ 0.015) \tag{14}$$

$$P_A = W_A \times R_A = (0.552,\ 0.249,\ 0.141,\ 0.057,\ 0.000) \tag{15}$$

Let the total evaluation vector be $L$ Combining the above leads to.

$$L = W_D \times \begin{bmatrix} P_M \\ P_O \\ P_A \end{bmatrix} = (0.521,\ 0.228,\ 0.161,\ 0.079,\ 0.010)$$

Finally, the evaluation results were summarized and weighted to obtain a final score of 77.51 for scheme $G_1$, 80.37 for scheme $G_2$, and 73.38 for scheme $G_3$, and the score results are shown in Fig 9. From the weighted score results, we can see that scheme $G_2$ has the highest score and is the optimal scheme, which verifies the correct letter of the evaluation study of agricultural intelligent robots based on the combination of KANO model, combined assignment and VIKOR method.

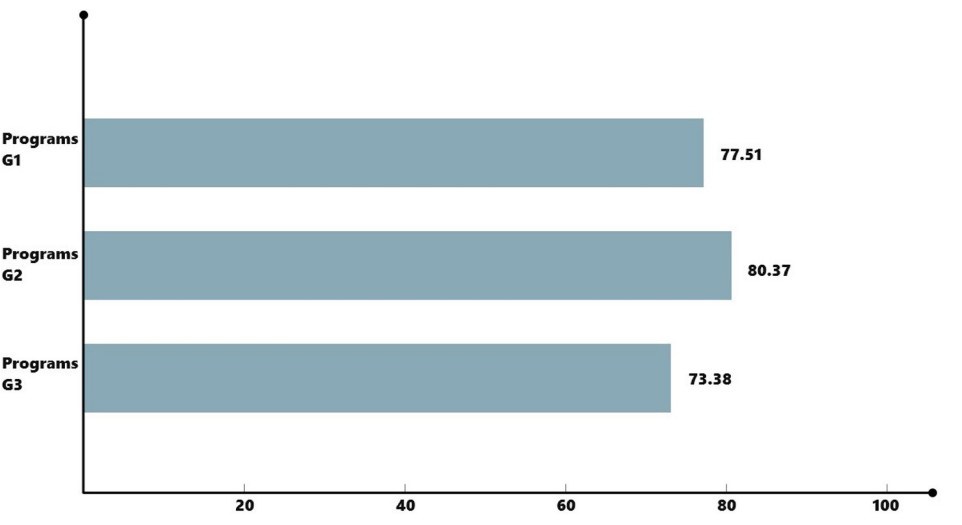

**Fig 9. Program score results.**

## 5.7 Elaboration and refinement of optimal solution

To further improve the satisfaction of the optimal solution for the users, the details of solution $G_2$ are designed as shown in Fig 10. The scenarios are used as shown in Fig 11. Information collection as the highest weighted factor in the index is applied to the three categories of soil information, crop health information, and environmental information acquisition. The retractable soil collector at the rear of the machine can obtain the PH value of the land and also the quality information of the cultivated soil. The mechanical arm is responsible for crop sample grabbing, and the mechanical claw part is set up with infrared as well as chlorophyll detector to collect crop chlorophyll information while grabbing. The sensor and camera function

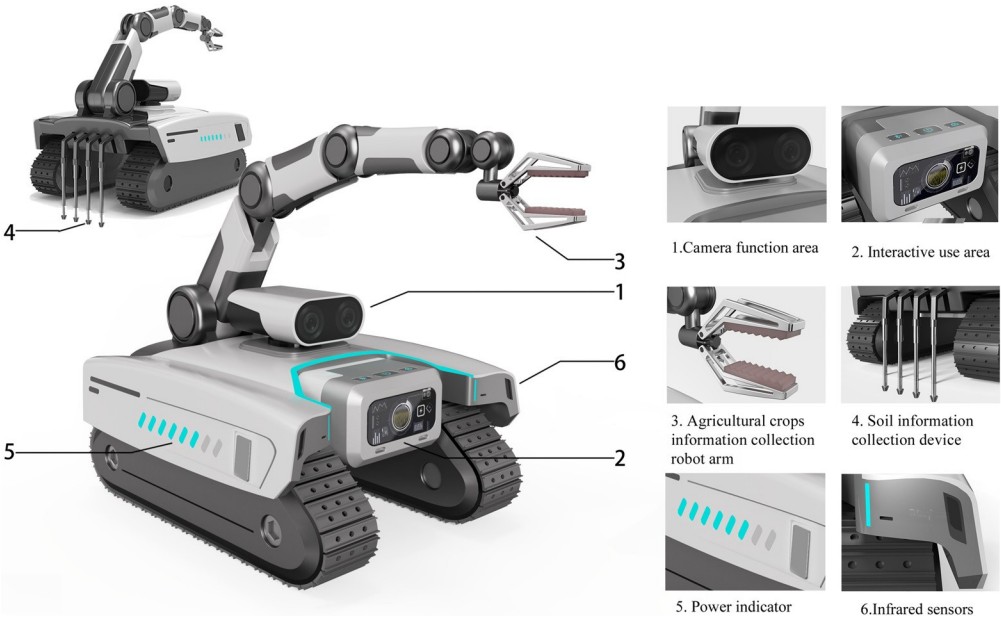

**Fig 10. Detail picture of the best solution.**

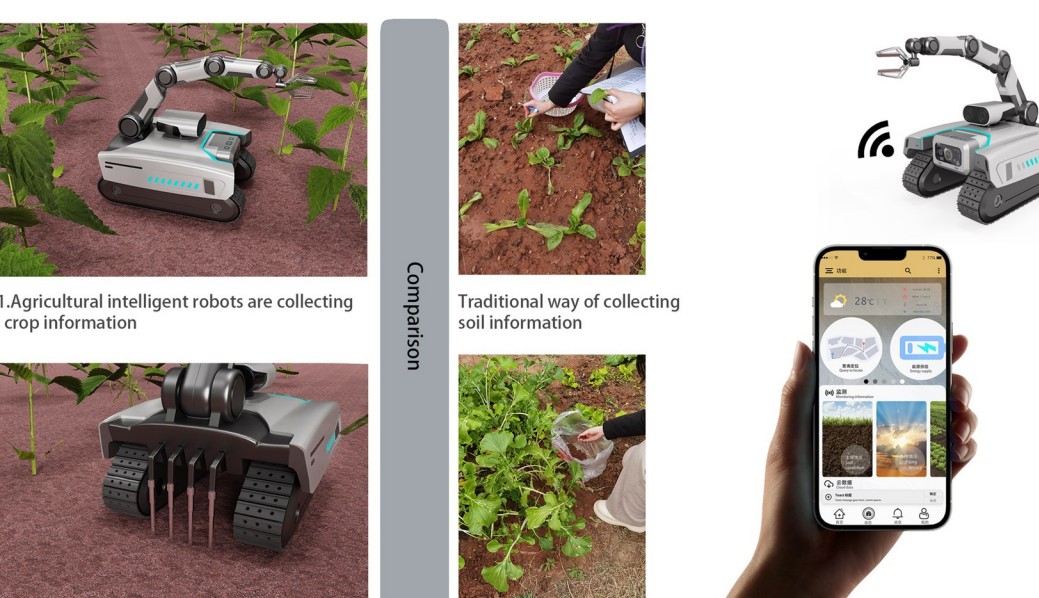

**Fig 11. Scenario usage diagram.**

can complete the task of collecting environmental information. The specific functional counterparts are shown in Table 14.

This optimal solution establish a data cloud platform based on the Internet of Things system, combine with coordinate positioning, complete real-time data storage and uploading functions, while data accumulation and analysis can also realize the management of agricultural safety timing and quantification, and improve the intelligent design of products. Combining ergonomic design of the control buttons and the layout of the operation screen, and adding a warning light design will make it more convenient for users to use. Remote control by cell phone APP through WIFI, Bluetooth, satellite positioning, Internet and other functions, and the ability to receive remote command signals during the work process, timely adjustment of the work plan as well as the function of the alarm system. Rotatable multi-view picture-taking features that can be used to monitor pests. The scientific design of agricultural intelligent robot's proportion, overall center of gravity, and chassis function can ensure the driving stability of the small mobile robot. The use of tracking design can ensure the adaptation of more terrain while driving. The overall shape is more innovative, with a sense of technology. The additional display of the robot can be used for the purpose of image interaction. Infrared sensors set at the front end can play an intelligent avoidance function. The material selection of non-ferrous metal combined with ABS material has advantages such as corrosion resistance.

**Table 14. Functional components for robot information collection.**

| Requirements | Components for functions |
|---|---|
| Agricultural crop health monitoring | Image Data Analysis/Chlorophyll Detectors |
| Environmental Information Monitoring | Light sensor/Humidity Sensor |
| Soil Information Monitoring | Pressure sensors/PH Sensors |
| Driving and avoidance | Infrared sensors/Image Data Analysis |

## 6. Discussion

The study conducted research and interviews on farming operations and farmers, and divided the collected raw information into five demand attributes using the Kano model, whereby a demand index system was established and used to complete the subsequent evaluation and preference of design solutions, reducing the drawback that the output solutions of designers are detached from the actual needs of users. The demand analysis for the farmer user group makes the design solution and evaluation study closer to the users themselves, so that the product will be more popular with the users when it is actually put into use and make the product better serve the farming operation.

In the process of assigning weights to evaluation indicators, the subjective assignment method AHP can use the empirical knowledge of decision makers to effectively control the weights of important indicators from the subjective level, and the entropy method can use the information entropy to carry real data information from the objective level. The use of the combination of subjective and objective assignment method has, to a certain extent, eliminated the one-sidedness and subjectivity of many scholars in the final assignment, and made the evaluation of demand indicators more scientific.

The VIKOR method is innovatively introduced in establishing the evaluation model of agricultural intelligent robot solutions. On the one hand, the VIKOR method considers both group utility maximization and individual regret minimization, and can also incorporate the subjective preferences of decision makers, which have higher ranking stability and credibility and enhances the scientifically of the decision model. On the other hand, the combination of the combined assignment method and the VIKOR method for scheme preference makes up for the disadvantages of the VIKOR method when used singularly, and provides a reference for precise quantitative analysis and accurate feedback of the design scheme.

The article gets the demand index of agricultural intelligent robots in order: Information collection> Remote operation>Data processing>Rational structure> Identification pests>Terrain adaptation>Fluent modeling>Intelligent avoidance> Drive driving>Preset work>Autonomous navigation>Button layout>Image interaction>Durable material>Anti-theft system>Color coordination. Three specific design options are proposed based on the requirement importance ranking. Among them, the requirement index of information collection is ranked first in importance, which indicates that the information collection in the field is the most important requirement for this type of intelligent robot for the purpose of real-time monitoring, and the designers should focus on this point. The importance of the color coordination index is the first to the last, indicating that for this type of intelligent robots in the color design tolerance is high, which can reduce the design cost appropriately. By analogy, the obtained requirement importance ranking can provide a reference for the design of future preferences related to agricultural intelligent robots under the IoT system.

The study of the article also has certain limitations, and although the advantages of the combined assignment and VIKOR method are proposed in the decision making part of the scheme, the calculation results are not compared with those of other multi-criteria decision making methods. In the next stage of this study, multiple multi-attribute decision methods will be compared and the results obtained will be used to select the optimal solution and specifically to verify the superiority of the combined assignment combined with the VIKOR method.

This study still has unreasonable data collection for calculation, although the use of methods such as the combination of subjective and objective enhances the scientifically of decision making to a certain extent, the number of raters selected in assigning weights is small, making the data limited. The follow-up study will expand the number of scores and consider using

instruments such as brain waves to verify the scoring results and increase the credibility and objectivity of the scoring data.

The user requirements collection part of the study regarding the Kano model, regardless of the fact that it has a large sample size, relies only on textual and verbal descriptions to describe the requirements issues when distributing the research questionnaire, which lacks sufficient objectivity. Considering that each individual in the user community has different perceptions of agricultural intelligent robots, subsequent studies should collect demand indicators from the perspective of perfect information after fully explaining to users about this type of robot.

## 7.Conclusions

In the context of the Internet of Things era, relying on new intelligent robots instead of manual work can not only improve industrial efficiency, but also promote the improvement of related technologies and systems. In existing research related to intelligent robots in agriculture, the needs of the user population is often ignored, resulting in a deviation of the final solution from user expectations. Therefore, the study takes the Kano model as the theoretical basis, establishes the evaluation index system of agricultural intelligent robot design based on the needs of farmers as a group, and determines 16 sub-benchmark evaluation indexes. The weights of each evaluation element were further obtained using the combined assignment method of AHP method and entropy method, which to a certain extent reduces the subjective influence in the decision-making process and at the same time makes full use of the experience of decision makers and plays a certain role as a reference aid. Finally, the combined assignment and VIKOR method of decision making are used to effectively improve the accuracy and comprehensiveness of the solution preference results. It provides a reference for converting the needs of farmers as a user group into design elements and putting them into the design and application of actual agricultural machinery in the future, and enhances the practical application value of subsequent products. It also provides a reference for the design evaluation of intelligent robots for agriculture under the IoT system.

## Supporting information

**S1 File.**
(DOCX)

## Acknowledgments

The authors would like to thank the School of Industrial Design, Hubei University of Technology for providing the research space and Ms. Tang Qian for her guidance on the article.

## Author Contributions

**Conceptualization:** Qian Tang, Ying-Wen Luo.

**Data curation:** Qian Tang, Ying-Wen Luo.

**Formal analysis:** Ying-Wen Luo, Xiao-Di Wu.

**Funding acquisition:** Ying-Wen Luo.

**Investigation:** Ying-Wen Luo.

**Methodology:** Ying-Wen Luo.

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
