## [Decision Letter · Decision Letter 0]

17 Nov 2022

PONE-D-22-27892Research on the evaluation method of agricultural intelligent robot design solutionsPLOS ONE

Dear Dr. Luo,

Thank you for submitting your manuscript to PLOS ONE. After careful consideration, we feel that it has merit but does not fully meet PLOS ONE’s publication criteria as it currently stands. Therefore, we invite you to submit a revised version of the manuscript that addresses the points raised during the review process.

We look forward to receiving your revised manuscript.

Kind regards,

Abdullah Hussein Abdullah Alamoodi, Ph.D.

Academic Editor

PLOS ONE

Journal Requirements:

2. Please provide additional details regarding participant consent to collect personal data, including email addresses, names, or phone numbers. In the Methods section, please ensure that you have specified how consent was obtained and how the study met relevant personal data and privacy laws. If data were collected anonymously, please include this information." 3) "Please provide additional details on the population sampled and its demographics. Please clarify how the survey was constructed, tested and distributed, and how participants were identified.

"The authors acknowledge funding from a grant from the Hubei Provincial Department of Education, grant numbers [21Q075],and the College of Industrial Design, Hubei University of Technology for providing the research space. Finally, the author thanks Qian Tang for her guidance on the article."

"Project source unit: 2021 Hubei Provincial Education Department Philosophy and Social Science Research Project [21Q075]"

"The author declares no conflicts of interest."

6. Please amend the manuscript submission data (via Edit Submission) to include author Xiao-Di Wu.

7. Please ensure that you refer to Figure 7 in your text as, if accepted, production will need this reference to link the reader to the figure.

8. Please remove your figures from within your manuscript file, leaving only the individual TIFF/EPS image files, uploaded separately. These will be automatically included in the reviewers’ PDF.

Reviewers' comments:

Reviewer's Responses to Questions

**Comments to the Author**

1. Is the manuscript technically sound, and do the data support the conclusions?

Reviewer #1: Yes

Reviewer #2: Yes

2. Has the statistical analysis been performed appropriately and rigorously? 

Reviewer #1: Yes

Reviewer #2: N/A

3. Have the authors made all data underlying the findings in their manuscript fully available?

Reviewer #1: Yes

Reviewer #2: Yes

4. Is the manuscript presented in an intelligible fashion and written in standard English?

Reviewer #1: Yes

Reviewer #2: Yes

5. Review Comments to the Author

Reviewer #1: The content of the article is consistent with the scientific area of the PLOS ONE. The subject raised by the authors is current and so far rarely noticed by other authors publishing in this area.

The issue described may in the future contribute to improving the efficiency of the automation of the transport process or agricultural intelligent robot ...

The paper has an original, scientific character, related to research on the evaluation method of agricultural intelligent robot design solutions. The authors focus on reducing subjectivity in program selection in the process of designing an agricultural robot and making more objective and rational decisions in order to increase the practicality and scientificity of the program. In addition, the authors correctly propose a new comprehensive evaluation method based on user requirements.

For a better clarification, please edit your paper as follows: 1. Extend the text of the manuscript (e.g. introduction or conclusion) with specific results in the world and Europe, - Improve the quality of the paper by presenting the results of publications by researchers and experts who are involved in this field and are registered in world databases (wos). These are e.g: Simulation of vertical vehicle non-stationary random vibrations considering various speeds, Analysis of stress and strain of fatigue specimens localised in the cross-sectional area of the gauge section testing on bi-axial fatigue machine loaded in the high-cycle fatigue region and Path planning optimization of six-degree-of-freedom robotic manipulators using evolutionary algorithms, thanks.

2. figure 10 should be contrasting and readable,

3. conclusions and future work should be extended to contain practical applications based on research described in this paper - expand references,

4. highlight the course of dependencies/relations in figure No. 1,

5. Unify font in table No: 14.

I recommend publishing the post after the proposed modifications.

Reviewer #2: Manuscript ID: PONE-D-22-27892

Review report for the paper “Research on the evaluation method of agricultural intelligent robot design solutions”.

Broadly speaking, this paper deals with the evaluation of agricultural intelligent robot design solutions. First, Kano model applied to classify the requirements and establishing an evaluation index system. Secondly, the combination of hierarchical analysis(AHP) and entropy weighting method is used to assign weights to the evaluation index system, calculate the weight value and importance ranking of each index, and carry out various program designs based on the ranking. Finally, the VIKOR method was applied to evaluate and rank the design solutions.

Some typo errors must be rectified. For example, ‘VILOR’ method. Thorough checking is needed.

Significance:

-The scientific content of this paper is correct.

-The results could be better presented. This would emphasize the quality of the presented work.

-The limits of the paper are mentioned but some of the points should be investigated.

Quality of presentation:

In eq.(1), the parameter’s range is taken as 0 ≤ ≤ 1. Why is only the decision preference coefficient =0.5 discussed? How do you fix this particular value? What does it signify? Would there be same results if either <0.5 or >0.5. Discuss few cases in each scenario.

Results – When a new evaluation framework is proposed, its result validation and comparisons of the results should be discussed. One can not leave it for future study as the results of the present study are not validated and compared. A sensitivity analysis and comparison are must to include.

Scientific soundness:

The subject addressed in this paper is relevant.

Interest to the readers:

In my opinion, the method of this paper seems to be interesting for the readership of the journal.

6. PLOS authors have the option to publish the peer review history of their article (what does this mean?). If published, this will include your full peer review and any attached files.

Reviewer #1: No

Reviewer #2: No

---

## [Author Response · Author response to Decision Letter 0]

29 Dec 2022

On behalf of my co-authors, we thank you very much for giving us an opportunity to revise our manuscript, we appreciate editor and reviewers very much for their positive and constructive comments and suggestions on our manuscript entitled “Research on the evaluation method of agricultural intelligent robot design solutions”.

Here are some notes on the changes required by the journal:

1. We have revised the article according to the style requirements of PLOS ONE.

2. The questionnaire was partially collected anonymously and based on the content of the article study, the population of agricultural workers was selected for the study. After conducting fieldwork, questionnaires were distributed to the relevant population in the area with the help of village managers, and according to the results of the returns, the population was 68% male and 32% female.

3. Funding Information is “Project source unit: 2021 Hubei Provincial Education Department Philosophy and Social Science Research Project [21Q075].”

4. We have removed the reference to grant information from the acknowledgements section.

5. We have added the statement "The author declares no competing interests" to the cover letter and submission system.

6. The details of the third author, Xiao-di Wu, are with the School of Industrial Design, Hubei University of Technology, 430068, Hubei, China. current address, 28 Nanli Road, Hongshan District, Wuhan, Hubei, China. Email for di1191495203@gmail.com。

7. We checked the text regarding Figure 7.

8. We removed all figures from the manuscript and only the captions of the images remained in the manuscript. And uploaded all the image files in a separate digital diagnostic tool of Pre-Assessment Analysis and Conversion Engine (PACE) as requested.

9. We edited the manuscript for language usage, spelling and grammar.

We have uploaded "Response to Reviewers" to explain in detail the changes made to the reviewers' comments.

To reviewer #1:

Thank you for the useful comments. The changes are highlighted in yellow in 'Revised Manuscript with Track Changes'. It is marked with yellow highlighting.

1. In the introduction section, we have added the publication results of experts in related fields to enhance the quality of the paper. 

2. Modified Figure 11 (original Figure 10) by adding annotations of usage scenarios to the figure, as well as comparing it with the traditional assignment method. 

3. In the conclusion, the role of the practical applications of the research in this paper in the future is emphasized.

4. Modified Figure 1 to emphasize the process of dependencies in the figure. 

5. Standardized the font of Table 14.

To reviewer #2:

Thank you for the useful comments. Changes are highlighted in pink in 'Revised Manuscript with Track Changes'. It is highlighted in pink.

1. Fixed some grammar and spelling errors.

2. In section 5.3, we add the relevant content about the decision coefficient β, discuss the influence of the value of the decision coefficient on the results under different situations, and determine the value of the decision coefficient of the article based on the publications of related studies and the research needs of this paper.

3. New chapter "5.6 Validation of evaluation results" is added to validate the research results of this paper using fuzzy evaluation method, and the validation results are passed.

---

## [Decision Letter · Decision Letter 1]

26 Jan 2023

Research on the evaluation method of agricultural intelligent robot design solutions

PONE-D-22-27892R1

Dear Dr. Luo,

We’re pleased to inform you that your manuscript has been judged scientifically suitable for publication and will be formally accepted for publication once it meets all outstanding technical requirements.

Kind regards,

Abdullah Hussein Abdullah Alamoodi, Ph.D.

Academic Editor

PLOS ONE

Additional Editor Comments (optional):

Reviewers' comments:

Reviewer's Responses to Questions

**Comments to the Author**

1. If the authors have adequately addressed your comments raised in a previous round of review and you feel that this manuscript is now acceptable for publication, you may indicate that here to bypass the “Comments to the Author” section, enter your conflict of interest statement in the “Confidential to Editor” section, and submit your "Accept" recommendation.

Reviewer #1: All comments have been addressed

Reviewer #2: All comments have been addressed

2. Is the manuscript technically sound, and do the data support the conclusions?

Reviewer #1: Yes

Reviewer #2: Yes

3. Has the statistical analysis been performed appropriately and rigorously? 

Reviewer #1: Yes

Reviewer #2: N/A

4. Have the authors made all data underlying the findings in their manuscript fully available?

Reviewer #1: Yes

Reviewer #2: Yes

5. Is the manuscript presented in an intelligible fashion and written in standard English?

Reviewer #1: Yes

Reviewer #2: Yes

6. Review Comments to the Author

Reviewer #1: The authors accepted the comments, I recommend the paper to be published, thanks.

The authors accepted the comments, I recommend the paper to be published, thanks.

Reviewer #2: The authors have addressed the point of my concern. I am happy with their corrections. Hence, I would like to recommend this manuscript to be published.

7. PLOS authors have the option to publish the peer review history of their article (what does this mean?). If published, this will include your full peer review and any attached files.

Reviewer #1: No

Reviewer #2: No

---

## [Editor Report · Acceptance letter]

1 Feb 2023

PONE-D-22-27892R1 

Research on the evaluation method of agricultural intelligent robot design solutions 

Dear Dr. Luo:

I'm pleased to inform you that your manuscript has been deemed suitable for publication in PLOS ONE. Congratulations! Your manuscript is now with our production department. 

Kind regards, 

on behalf of

Dr. Abdullah Hussein Abdullah Alamoodi 

Academic Editor

PLOS ONE